# Structural Entities Associated with Different Lipid Phases of Plant Thylakoid Membranes—Selective Susceptibilities to Different Lipases and Proteases

**DOI:** 10.3390/cells11172681

**Published:** 2022-08-28

**Authors:** Ondřej Dlouhý, Václav Karlický, Uroš Javornik, Irena Kurasová, Ottó Zsiros, Primož Šket, Sai Divya Kanna, Kinga Böde, Kristýna Večeřová, Otmar Urban, Edward S. Gasanoff, Janez Plavec, Vladimír Špunda, Bettina Ughy, Győző Garab

**Affiliations:** 1Group of Biophysics, Department of Physics, Faculty of Science, University of Ostrava, 710 00 Ostrava, Czech Republic; 2Laboratory of Ecological Plant Physiology, Global Change Research Institute of the Czech Academy of Sciences, 603 00 Brno, Czech Republic; 3Slovenian NMR Center, National Institute of Chemistry, SI-1000 Ljubljana, Slovenia; 4Photosynthetic Membranes Group, Institute of Plant Biology, Biological Research Centre, Eötvös Loránd Research Network, 6726 Szeged, Hungary; 5Doctoral School of Biology, University of Szeged, 6726 Szeged, Hungary; 6Belozersky Institute of Physico-Chemical Biology, M.V. Lomonosov Moscow State University, 119991 Moscow, Russia; 7STEM Program, Science Department, Chaoyang KaiWen Academy, Beijing 100018, China; 8EN-FIST Center of Excellence, SI-1000 Ljubljana, Slovenia; 9Faculty of Chemistry and Chemical Technology, University of Ljubljana, SI-1000 Ljubljana, Slovenia

**Keywords:** ^31^P-NMR spectroscopy, lipid polymorphism, lipocalins, membrane fusion, membrane models, non-bilayer lipids, plastoglobuli, structural and functional plasticity, thylakoid membrane

## Abstract

It is well established that plant thylakoid membranes (TMs), in addition to a bilayer, contain two isotropic lipid phases and an inverted hexagonal (H_II_) phase. To elucidate the origin of non-bilayer lipid phases, we recorded the ^31^P-NMR spectra of isolated spinach plastoglobuli and TMs and tested their susceptibilities to lipases and proteases; the structural and functional characteristics of TMs were monitored using biophysical techniques and CN-PAGE. Phospholipase-A1 gradually destroyed all ^31^P-NMR-detectable lipid phases of isolated TMs, but the weak signal of isolated plastoglobuli was not affected. Parallel with the destabilization of their lamellar phase, TMs lost their impermeability; other effects, mainly on Photosystem-II, lagged behind the destruction of the original phases. Wheat-germ lipase selectively eliminated the isotropic phases but exerted little or no effect on the structural and functional parameters of TMs—indicating that the isotropic phases are located outside the protein-rich regions and might be involved in membrane fusion. Trypsin and Proteinase K selectively suppressed the H_II_ phase—suggesting that a large fraction of TM lipids encapsulate stroma-side proteins or polypeptides. We conclude that—in line with the Dynamic Exchange Model—the non-bilayer lipid phases of TMs are found in subdomains separated from but interconnected with the bilayer accommodating the main components of the photosynthetic machinery.

## 1. Introduction

In oxygenic photosynthetic organisms, light reactions occur in flattened membrane vesicles, the thylakoid membranes (TMs). In chloroplasts, the multilamellar TM system resides in the aqueous matrix, called the stroma. Plant TMs are differentiated into a stacked or appressed region, the granum, and an unstacked or non-appressed region—the stroma lamellae; they form an extended intricate 3D TM network with a single continuous membrane, which encloses an interconnected contiguous interior aqueous phase, the lumen [1,2].

TMs contain membrane-intrinsic pigment-protein complexes (PPCs), namely, the two photosystems, PSII and PSI, along with their peripheral light-harvesting antenna complexes, LHCII and LHCI, respectively; they are assembled into large PSII–LHCII and PSI–LHCI supercomplexes, which are enriched in the grana and the stroma lamellae, respectively. The membranes also accommodate the cytochrome b_6_f complex and the F_o_ subunit of the ATP synthase, as well as small lipophilic molecules such as plastoquinone, which participates in electron and proton transfer processes. Further constituents of the energy-converting machinery are found in the inner and outer aqueous phases of the TMs, i.e., the water-splitting enzyme, the ferredoxin-NADP^+^ reductase (FNR), the F_1_ subunit of the ATP synthase and the mobile electron carriers plastocyanin and ferredoxin. Further important water-soluble protein constituents include the luminal- and stromal-side enzymes of the xanthophyll cycle, VDE (violaxanthin de-epoxidase) and ZE (zeaxanthin epoxidase), respectively [3], and the luminal plastid lipocalin (LCNP) [4]. In general, both the interthylakoidal space and the lumen are crowded with proteins and polypeptides [5,6].

The build-up of the proton motive force and its utilization for ATP synthesis, according to the chemiosmotic mechanism [7], is warranted by the organization of the TM lipids into bilayer structures, which are essentially impermeable to water and different ions. Hence, the functional state of the bulk TM lipids, representing about 60% of all lipids [8], must be bilayer—in accordance with the “standard” fluid-mosaic membrane model [9]. However, it is well known that the major lipid species in TMs, which constitutes about half of the total lipid contents of TMs, MGDG (monogalactosyldiacylglycerol), is a non-bilayer- or non-lamella-forming lipid [10]. These types of lipids, because of their conical shapes [11], are not capable to self-assemble into bilayers under physiologically relevant conditions; they prefer to form different non-lamellar or non-bilayer lipid phases, such as the inverted hexagonal (H_II_), isotropic, and cubic phases. The rest of the TM lipids—digalactosyldiacylglycerol (DGDG; ~25–30%), sulfoquinovosyldiacylglycerol (SQDG; ~10–15%), and phosphatidylglycerol (PG; ~10–15%) [12,13,14]—possess cylindrical shapes and are capable to spontaneously assemble into bilayer structures. 

Because of the abundance of MGDG, bulk TM lipids have strong non-bilayer propensity; this can lead to the formation of non-lamellar lipid phases [10,15,16]. In fact, fully functional isolated intact TMs have been shown to contain non-bilayer lipid phases, as revealed via ^31^P-NMR spectroscopy; the polymorphism of lipid phases has also been confirmed by time-resolved fluorescence spectroscopy using Merocyanine-540-stained TMs [17,18,19]. These features—the dominance of non-bilayer lipid species and the polymorphism of lipid phases—are not unique to TMs but also hold true for the other main energy-converting membrane, the inner mitochondrial membrane (IMM) [20,21].

The abundance of non-bilayer lipids in energy-converting membranes is difficult to explain within the framework of the generally accepted, “standard” fluid-mosaic membrane model, because it does not consider the presence of non-bilayer lipids [9,22]. The fact that all biological membranes contain non-bilayer lipid species (cf. [23,24]) motivated the construction of membrane models that go beyond the “standard” model. The Lateral Pressure Model (LPM) postulates that the presence of non-bilayer lipids in the bilayer membrane is essential to “keep the [proteins] in a functional state” via increasing the lateral pressure in the hydrophobic region of the bilayer membranes and decreasing it in the region of lipid headgroups [25]. The Flexible Surface Model (FSM) predicts that “the non-lamellar-forming tendency of the membrane lipids modulates the protein energetics” [26]. These models emphasize that the presence of non-bilayer lipids in the bilayer membrane enhances the structural flexibility of membranes and that the frustrated state of the bilayer lends dynamics to bilayer-embedded proteins (cf. also [27]). Similar mechanisms have been proposed to explain the structural and functional plasticity of LHCII in MGDG-containing model systems [28,29]. In addition, according to a recent modification of the “standard” model, the shape of membranes—influenced by the presence of non-bilayer lipids—was proposed to modulate protein properties [30]. However, it must be pointed out that none of the above models permit bilayer and non-bilayer phases to coexist in TMs nor do they explain the remarkable structural dynamics of the lipid phases [17,18,19,31]. These features can be explained within the frameworks of the Dynamic Exchange Model (DEM) [32,33]. DEM is based on the Janus face of lipid mixtures of high non-bilayer propensity—their ability to enter (be forced into) the bilayer and to segregate from it; this dynamic exchange of lipids between the different phases has been proposed to self-regulate the homeostasis of TMs [32]. Recently, it has been hypothesized that plastoglobuli, which bud from the outer leaflet of TMs [34], might also participate in maintaining a constant lipid:protein ratio via a mechanism of dynamic exchange of lipids between TMs and plastoglobuli [35].

In our earlier studies, we established correlations among the variations in the temperature, pH, and osmotic and ionic strengths of the medium as well as the activity of VDE and the phase behavior of the bulk lipid molecules [31,36,37]; we also clarified that isolated grana and stroma lamellae contain all the three non-bilayer phases [38]. Analyses of our structural and functional data strongly suggested that the isotropic phases originate from membrane fusions and junctions [39] and from VDE: lipid assemblies [31]. (Note that in model membranes, VDE: lipid structures form an H_II_ phase [40].) Further, we proposed—also based on literature data [41,42]—that the excess lipids that are extruded from the bilayer membrane assemble into an H_II_ phase. However, these assignments should be substantiated, and further experimental data are needed to explore the origin and the structural and physiological significance of the different lipid phases in TMs. In addition, the role of plastoglobuli in the polymorphic phase behavior of TMs has not been clarified.

In this work, to obtain information about the identity of structural entities encompassing the different ^31^P-NMR-detectable non-bilayer lipid phases of isolated TMs, we investigated their sensitivity to different lipases and proteases; moreover, under the same experimental conditions, we also monitored how these treatments affected the structure and function of the photosynthetic machinery. The obtained data clarified that the non-bilayer phases are located in the subdomains of the TMs that are separated from but interconnected with the protein-rich bilayer membrane accommodating the membrane-intrinsic proteins of the photosynthetic machinery. Our data obtained from isolated plastoglobuli—a weak and very sharp, phospholipase-insensitive ^31^P-NMR signal—rule out their sizable contribution to the lipid polymorphism of TMs. In addition, our data on proteases revealed that an H_II_ phase arises from a large population of lipid molecules that surround or encapsulate stromal-side protein subunits or polypeptides.

## 2. Materials and Methods

### 2.1. Isolation of Thylakoid Membranes

Spinach TMs were isolated as previously described [31]. Spinach leaves were purchased at a local supermarket and stored at 4 °C in the dark for 1–3 days before use. Leaves were homogenized in buffer A (50 mM Tricine, 5 mM MgCl_2_, 5 mM KCl, and 400 mM sorbitol; pH 7.5). The suspension was filtered using four layers of cheese cloth and centrifuged for 2 min at 400× *g*; then, the supernatant was centrifuged again for 10 min at 6000× *g*. The pellet containing chloroplasts was osmotically shocked for 10 s using hypotonic buffer B (50 mM Tricine, 5 mM MgCl_2_, and 5 mM KCl; pH 7.5) followed by the addition of buffer 2A, with doubled osmotic strength (800 mM sorbitol, 50 mM Tricine, 5 mM MgCl_2_, and 5 mM KCl; pH 7.5). The disrupted chloroplasts were centrifuged for 10 min at 6500× *g*. The pellet containing TMs was finally resuspended in buffer A. The chlorophyll (Chl) content of the TMs was determined according to [43]. The isolation steps were performed on ice under dim light.

### 2.2. Isolation of Plastoglobuli

Plastoglobuli were isolated according to [44] with some modifications by Zach Adam (Weizmann Institute of Science, Rehovot, Israel). Plant material was homogenized in AGB buffer (20 mM HEPES-KOH (pH 8), 2 M sorbitol, 0.5 M EDTA, 1 M MgCl_2_, and 0.5 M EGTA) with 0.5 g/L BSA in a plant material:buffer volume ratio of 1:10. The suspension was filtered through a miracloth and centrifuged at 1500× *g* for 5 min, and the pellet was resuspended in AGB buffer. The lysate was loaded on PBF percoll gradient (3 mL of 80% pillow + 7 mL of 40% step gradient in 15 mL corex tubes) and continuously centrifuged at 1500× *g* for 15 min. The first band was collected, resuspended in a medium containing 50 mM HEPES and 0.33 M sorbitol, and centrifuged at 3000× *g* for 10 min. The pellet containing TMs was resuspended in medium containing 50 mM Tricine-KOH (pH 7.5), 2 mM EDTA, and 0.2 mM dithiotreitol (TrE medium) with added proteinase inhibitor. Chlorophyll concentration was determined according to [43].

The isolated TMs were suspended in TrE medium in a 1:600 ratio and centrifuged at 8000× *g* for 10 min; the pellet was resuspended in 45% sucrose/TrE to a final chlorophyll concentration of 1–3 mg Chl/mL), homogenized in a 15 mL Potter homogenizer with 20 strokes, and sonicated 3 × 5 s with 5 s pauses in between. Approximately 2.5–3 mL of the homogenate was poured into UltraClear tubes SW28 and carefully overlayed with the sucrose/TrE solutions in the following order: 3 mL of 38% sucrose/TrE, 2 mL of 20% sucrose/TrE, 2 mL of 15% sucrose/TrE, and 3 mL of 5% sucrose/TrE; this was followed by overnight centrifugation at 100,000× *g*. The yellow band on top, containing plastoglobuli, and the bottom band, containing TMs, were collected. The isolated plastoglobuli were concentrated using an Amicon Ultra 4 centrifugal filter. All procedures were performed under dim green light at 5 °C.

### 2.3. Enzymatic Treatments 

Isolated TMs were treated with (i) phospholipase A1 (PLA1; L3295; Sigma-Aldrich, Burlington, MA, USA)—a lipase cutting the *sn-1* position of phospholipids—and (ii) wheat germ lipase (WGL)—a substrate non-specific (Sigma-Aldrich website), general tri-, di-, and monoglyceride hydrolase—lipase (L3001; Sigma-Aldrich, Burlington, MA, USA) to digest all the classes of lipids contained in the samples. Stock solutions with activities of 4000 and 1000 U mL^−1^, respectively, were dissolved in demineralized water; if not indicated otherwise, 5 µL of lipase stock per 1 mL of sample was added to a final activity of 2 and 5 U mL^−1^, respectively, and thoroughly mixed. All samples were incubated at 5 °C for the required times before measurements.

For Trypsin (T8003; Trypsin from bovine pancreas; Sigma-Aldrich, Burlington, MA, USA) treatment, a stock of 300 mg mL^−1^ was dissolved in demineralized water, and 33 µL per mL of sample was added for a total concentration of 10 mg mL^−1^. The suspension was thoroughly mixed and kept at 5 °C until the start of the measurements. Proteinase K solution (P8107S; New England BioLabs, Ipswich, MA, USA), containing 50% glycerol, was used at a final activity of 16 U mL^−1^. The Proteinase-K-treatment results were compared to control samples containing the same amount of glycerol as the proteinase-treated samples. The Proteinase-K-treated and control samples were incubated for 30 min at room temperature before they were cooled down to 5 °C and used.

All enzymatic treatments were performed on samples with Chl contents between 8 and 10 mg mL^−1^; if not indicated otherwise, they were conducted at 5 °C. This was also true for all the optical spectroscopic and biochemical measurements, during which aliquots were taken from the control and treated samples.

### 2.4. ^31^P-NMR Measurements

^31^P-NMR spectrometry was performed similarly as in our previous study [38]. ^31^P-NMR spectra were acquired with an Avance Neo 600 MHz NMR spectrometer (Bruker, Billerica, MA, USA) with BBFO SmartProbe tuned at the phosphorus frequency. Approximately 700 µL of sample was loaded into 5 mm NMR tubes. The TMs’ very high density did not allow any magnetic orientation of the sample, as we showed before [17]. For the acquisition of the spectra, 40° RF pulses with an inter-pulse time of 0.5 s with no ^1^H-decoupling were applied. An 85% solution of H_3_PO_4_ in water was used as an external chemical-shift reference.

In saturation transfer (ST) experiments, we applied RF pulses with low power at the selected frequency for 0.3 s, followed by 40° pulses with an acquisition time of 0.2 s with a repetition time of 0.5 s. The pre-saturation pulse intensity was adjusted regarding the saturated peak. The field strengths of RF pulses for pre-saturation were as follows: lamellar and H_II_ phases, 80 Hz; isotropic phases, 40 Hz. ^31^P-NMR spectra were normalized according to the Chl contents and the number of scans and then averaged from several samples.

The processing of the ^31^P-NMR spectra was performed using TopSpin (Bruker, Billerica, MA, USA) software, and mathematical deconvolution was performed in DMfit software (Dominique Massiot, Orléans, France) [45]. Individual lipid phase shapes were overlapped using their typical spectral distributions [46,47]. The figures were plotted using MatLAB R2018a (MathWorks, Inc., Portola Valley, CA, USA) with an implemented Spectr-O-Matic toolbox for the analysis of spectroscopic data (Dr. Petar H. Lambrev, Szeged, Hungary).

### 2.5. Electrochromic Shift Absorbance Transients (ΔA_515_)

The electrochromic shift (ECS) absorbance transients (ΔA_515_) were measured using a Joliot-type kinetic spectrometer (JTS-100; BioLogic, Grenoble, France) on dark-adapted isolated TMs with 20 µg mL^−1^ Chl content in a 1 cm × 1 cm cuvette. The detectors were protected against stray light from the excitation beam using a BG-39 bandpass filter, and a 520 or 546 nm interference filter was positioned after the measuring flashing beam from a LED source. An LF1 xenon-flash light source (Hamamatsu Photonics, Hamamatsu, Japan) providing single-turnover saturating flashes at 90° with respect to the measuring beam was used to activate the sample. Averages of five 0.15 s^−1^ repetition rate kinetic traces were acquired.

### 2.6. Circular-Dichroism Spectroscopy

A J-815 spectropolarimeter (Jasco, Tokyo, Japan) was used to acquire circular dichroism (CD) spectra of isolated TMs in the wavelength range between 350 and 750 nm. The spectra were acquired in 0.5 nm steps, integrated for 1 s, using bandpass of 2 nm and a speed of scan 100 nm min^−1^ in a cuvette with 1 cm optical pathlength. The Chl concentration of the sample was adjusted to 20 µg mL^−1^. The temperature in the sample chamber was set at 5 °C, controlled by a Peltier sample holder (Jasco, Tokyo, Japan). 

### 2.7. Chl-a Fluorescence Spectroscopy at 77 K

Fluorolog 3–22 spectrofluorometer (Horiba Jobin Yvon, Paris, France) with a Dewar-type optical cryostat was used to measure 77 K Chl-*a* fluorescence spectra of isolated TMs. To avoid reabsorption, TM suspensions were diluted to 5 µg Chl mL^−1^ in a 2 mm capillary tube. For emission spectra, the wavelength of the preferential excitation of Chl-*b* at 475 nm was used, with slit widths of 3 and 2 nm for excitation and emission monochromators, respectively. The spectral sensitivity of the detector was used as an emission-spectrum correction. Spectra were normalized to the PSI emission peak.

### 2.8. Fast Chl-a Fluorescence (OJIP) Transients

Isolated TMs were dark-adapted, diluted to 50 µg mL^−1^ Chl content, and measured in a 1 cm optical pathlength cuvette, using a FluorPen FP 100 fluorometer (Photon Systems Instruments, Drásov, Czech Republic), equipped with exciting short-wavelength LEDs of 3000 µmol photons m^−2^ s^−1^. Fast Chl-*a* fluorescence (OJIP) transient experiments were also performed in parallel with the ^31^P-NMR spectroscopy measurements on the same batch and incubation times using a Handy-PEA Chl-*a* fluorimeter (Hansatech Instruments Ltd., Pentney, United Kingdom), in which the sample was illuminated for 3 s with red light at a photon flux density of 3500 μmol photons m^−2^ s^−1^. The two sets of experiments followed very similar trends. The F_v_/F_m_ ratio was determined as (F_m_ − F_o_)/F_m_, with F_o_ and F_m_ being the minimum and the maximum levels of fluorescence, respectively.

### 2.9. CN-PAGE

A native electrophoresis of PPCs was carried out as previously described [48] with some modifications [49]. The mild solubilization of isolated TMs was performed by adding n-dodecyl-α-D-maltoside (α-DM) in the detergent to a total Chls ratio of 35:1 (10 min at 4 °C), followed by centrifugation at 4 °C, 21,000× *g* for 10 min. The aliquots (10 µg total Chls) of supernatant containing solubilized TMs were loaded into gel (4.5–11.5% linear gradient polyacrylamide gel), and the PPCs were separated at 4 °C in the dark with regime of increasing voltage from 75 to 200 V and a duration of 3.5 h [50]. Gels were scanned using a ChemiDoc MP gel imager (Bio-Rad Laboratories, Hercules, CA, USA) in transmitting white light and under blue excitation light to capture room-temperature Chl-*a* fluorescence.

## 3. Results and Discussion

As previously shown [19,31] and demonstrated in Appendix A, freshly isolated intact spinach TMs contain four well-discernible lipid phases: a lamellar (bilayer) phase (L), with a characteristic broad band with a peak at around δ_P_ −10 ppm and extending towards the low-field side of the ^31^P-NMR spectrum; two isotropic phases (I_1_ and I_2_), showing sharp peaks at around δ_P_ 2 and 4 ppm, respectively; and an inverted hexagonal phase (H_II_), with a maximum at δ_P_ ~25–30 ppm and shoulder on the high-field side. As reflected by the integrated areas of the component spectra, non-bilayer phases comprise a large fraction of the bulk lipid population (Appendix A). Their susceptibility to lipases and proteases might carry information about the structural entities associated with these phases.

### 3.1. Effects of Lipases

Two types of lipases were used and characterized in detail regarding their effects on the lipid polymorphism of TMs: Phospholipase A1 (PLA1), which selectively cleaves off the fatty acid on the *sn-1* position from the glycerol backbone [51], and Wheat-Germ Lipase (WGL), which is a substrate non-specific/promiscuous lipid hydrolase [52].

#### 3.1.1. Phospholipase A1

As shown in Figure 1, PLA1 treatment (2 U mL^−1^) rapidly (in 15 min) diminished all the lipid phases of TMs. Parallel with the loss of the characteristic ^31^P-NMR signals, a broad symmetric band emerged, peaking at around δ_P_ 15 ppm—indicating the restructuring of the lipid phases as a result of the presence of PG breakdown products. In the following 30–60 min, the remaining signals arising from H_II_ and the lamellar phase were completely lost (Figure 1 RHS inset), and the spectra were dominated by newly emerging isotropic bands that, after 60 min, peaked at around δ_P_ 1.5 and 3.3 ppm (Figure 1 LHS inset). 

Upon applying a higher concentration of PLA1 (24 U mL^−1^), the changes occurred faster; the H_II_ and the lamellar phases were destroyed by the end of the 5 min acquisition time, and a broad band, similar to that seen with 15 min 2 U mL^−1^ treatment, was formed with peak position at around δ_P_ 10 ppm (Appendix A). The further incubation of the sample in the presence of PLA1 led to the emergence of intense isotropic peaks at around δ_P_ 1.5 and 3.3 ppm. These high intensity signals might contain contributions from PG breakdown products from the annular and non-annular phases, i.e., arising from the gradual digestion of PG molecules originally found in the shell of the supercomplexes and/or bound to different proteins [53]. However, the magnitude of these contributions did not exceed 20% of bulk-phase PG, as judged from the <20% increase in the integrated areas of the spectra between δ_P_ 50 and −30 ppm (data not shown). 

Parallel with the weakening of the lamellar phase, the 15 min treatment of TMs with 2 U mL^−1^ PLA1 dramatically accelerated the decay of the flash-induced ΔA_515_, indicating that the bilayer membrane lost its impermeability (Figure 2). This observation is in harmony with the close correlation between the enhanced membrane permeability and the diminishment in the lamellar phase [37]. The further digestion of the membranes led to the loss of the initial amplitude measured at 1 ms after flash excitation (Figure 2 inset); this can be ascribed to the gradually increasing permeability of membranes, but as shown below, it can also be the consequence of reduced photochemical activity of TMs.

Treating TMs for ≤60 min with 2 U mL^−1^ induced no clearly discernible changes in the 77 K fluorescence emission spectra, showing that losses of the characteristic lipid phases of TMs induce very little effect on the excitation-energy distribution among the different PPCs (Figure 3). This can be explained by the notion that most of the excitation-energy transfer events occur inside the tightly packed PSII and PSI supercomplexes [54,55]. The fact that only longer PLA1 treatments lead to significant changes in the 77 K fluorescence emission spectra allows us to conclude that variations in the polymorphism of TM lipid phases exert no significant effects on the supercomplexes, the integrity of which is indeed largely retained even after detergent solubilization and crystallization [56,57]. Longer treatments, however, suppressed the PSII-associated 685 and 695 nm bands, relative to the long-wavelength bands of PSI-LHCI, and led to the emergence of intense emission at around 700 nm, which was attributed to LHCII aggregates [58]). (For the stepwise increase of F700/F695, see the inset in Figure 3.) The observed long-term effects of PLA1 most likely indicate the digestion of PG molecules of LHCII trimers [59] and PSII dimers [60]. 

To monitor the effects of PLA1 on the functional activity of PSII, we recorded fast Chl-*a* fluorescence (OJIP) transients at different intervals following lipase treatment (Figure 4). In accordance with the 77 K fluorescence emission spectroscopy data (Figure 3), short-term PLA1 treatments (≤30 min) exerted only minor effects on OJIP transients. On the other hand, longer-term incubations gradually diminished the activity of PSII (Figure 4). Prolonged lipase treatment caused an increase in the minimum fluorescence F_o_ level (Figure 4b), indicating the detachment of the LHCII complexes from the PSII supercomplexes. At the same time, PLA1 treatment gradually decreased the F_v_/F_m_ ratio (Figure 4c), a parameter that is characteristic of the photochemical activity of PSII and light-induced conformational changes/dielectric relaxation processes [61]. In general, the low susceptibility of PSII to short-term lipase treatments corroborates our conclusion that the functional activity of PSII is essentially insensitive to the lipid-phase behavior of TMs. At the same time, we note that our findings, concerning the gradual loss of PSII activity after longer PLA1 treatments, are consistent with similar data obtained for different phospholipases [53,60,62,63,64]. (N.B.: Because of the instability of the bilayer phase at higher temperatures, our PLA1 treatments were performed at 5 °C, which evidently slowed down the rate of activity of this lipase.)

The PLA1 treatment of TMs also affected the distribution and aggregation of PPCs, as reflected in CN-PAGE experiments (Figure 5). These data revealed that the lipase treatment mainly affected the PSII–LHCII supercomplexes, an effect which may be responsible for the diminished 77 K PSII fluorescence emission bands relative to that of PSI and the decreased PSII activity (cf. Figure 3 and Figure 4, respectively). 

Interestingly, PLA1 treatment exerted no noticeable effects on the CD spectra (Appendix A), indicating that the PPCs retained their native macro-organization—despite the complete loss of the bilayer phase. This observation may be explained by the inherently high protein content of TMs, which forms large arrays and chiral macrodomains with long-range order of the chromophores [65,66]. Extended arrays with long-range chiral order can also be formed by largely delipidated isolated LHCII proteins [67].

In summary, our data obtained from PLA1 (i) demonstrated that as expected, all lipid phases were sensitive to the mild hydrolysis of PG molecules, (ii) confirmed that the destruction of the bilayer phase instigated a rapid loss of the impermeability of TMs, and (iii) showed that the digestion of non-annular PG molecules, buried inside different PPCs, lagged behind the destabilization of the bulk lipid phases.

#### 3.1.2. Wheat-Germ Lipase

In contrast to PLA1, WGL, at 5 U mL^−1^, had no noticeable effects on the L and H_II_ phases. Instead, it selectively suppressed the isotropic phases and generated a broad signal between δ_P_ 30 and −10 ppm, indicating the formation of a largely immobilized lipid phase containing breakdown products. The addition of 20 U mL^−1^ WGL further suppressed the isotropic phases and diminished the L and H_II_ phases; at 100 U mL^−1^ of WGL, all phases were eliminated (Figure 6). These data are in good agreement with our previous results on isolated granum and stroma sub-chloroplast TM particles [38,39]. It is important to note that 5 U mL^−1^ WGL, even over long incubation times, did not disturb the structure and function of TMs; it did not increase the permeability of the membranes and exerted no effects on the excitation-energy distribution in the TMs, on the activity of PSII, nor the pattern and macro-organization of PPCs (Appendix A). 

These data show that the primary targets of WGL are to be found outside the bilayer and the protein-rich areas. These subdomains of the TMs evidently contain a large number of highly mobile lipid molecules—as indicated by the high-intensity sharp isotropic ^31^P-NMR spectral components, I_1_, I_2_, and I_i_, which comprise about 20% of the bulk lipid molecules (cf. Appendix A). The involvement of the highly curved margin of grana can most likely be ruled out as a structural unit responsible for the isotropic peaks, because the margin is occupied by CURT proteins [68]. (In addition, as already discussed in [17], the area occupied by the margin is very small compared with the area of the flat membrane regions.)

Structural entities containing lipids with rapid isotropic motion might be found in the junction of the granum and stroma TMs and in regions where adjacent stroma lamellae are fused together [2]. Non-bilayer lipid structures—such as stalks and fusion pores—have been shown to play key roles in membrane fusion [69,70,71]; furthermore, TM lipids readily form stalks [15]. In addition, the activity of VDE depends on the presence of non-bilayer lipid phases [3,31,40]; similar lipid phases might be formed around other water-soluble lipocalin-like proteins or lipocalins, such as stromal-side zeaxanthin-epoxidase [72] and lumenal plastid lipocalin LCNP [4]. 

### 3.2. Contributions from Plastoglobuli

Since lipid trafficking between TMs and plastoglobuli has been shown to play key roles in thylakoid functions from biogenesis to senescence [34,35], it is important to clarify if and to what extent plastoglobuli contribute to the dynamic polymorphic phase behavior of TMs. To this end, we compared the ^31^P-NMR spectra of plastoglobuli and TMs, isolated from approximately equal amounts of spinach leaves. As shown in Figure 7a, plastoglobuli exhibited a very weak and sharp ^31^P-NMR signal, with an integrated area more than an order of magnitude weaker than TMs’ isotropic region. The peak position of plastoglobuli, 2.04 ± 0.32 ppm (*n* = 6), was close to that of the I_1_ phase, and its half-bandwidth was much smaller and more in line with inorganic phosphates, which are expected to resonate in the same frequency range. It thus appeared that the signal observed in these spectra originated from some small phosphorus-containing molecules with rapid isotropic mobility, rather than from PG. Indeed, as shown in Figure 7b, the ^31^P-NMR signal of plastoglobuli was essentially insensitive to PLA1 treatment. The apparent absence of a sizeable contribution of plastoglobuli to the ^31^P-NMR signal of isolated TMs is consistent with the data showing that these lipid droplets contain only trace levels of TM lipids [73]. 

### 3.3. Effects of Proteases

The integrity and macro-organization of TMs depend largely on the interactions among surface-exposed proteins [74,75,76]. To test the putative effects of such interactions on the polymorphic phase behavior of TMs, we performed experiments using serine proteases Trypsin and Proteinase K. Trypsin favors basic residues such as lysine and arginine, and Proteinase K hydrolyzes a variety of peptide bonds. These proteases are routinely used in investigations of protein topology in TMs, since they selectively hydrolyze peptide bonds exposed to the stromal side of isolated intact TMs, while—with moderate digestions—they have no access to the luminal membrane surface [77,78,79,80]. In fact, Trypsin digestion of isolated plant TMs has been shown to eliminate the stacking of adjacent granum membranes [81]. 

#### 3.3.1. Trypsin

As shown in Figure 8, ^31^P-NMR spectroscopy revealed that Trypsin treatment of TMs induced a marked diminishment in the H_II_ phase, with little or no effect on the lamellar phase and with only some minor alterations in the I_1_ and I_2_ isotropic peaks. A slight change in the shape of the spectral signature compared with that of the L phase was also observed throughout the chemical-shift range (Figure 8 inset), suggesting the presence of a new signal originating from PG-containing breakdown products. The presence of this signal was confirmed with saturation-transfer experiments (Appendix A), which revealed the emergence of a new largely symmetric signal centered at δ_P_ around 15 ppm that contributed to the observed signal between δ_P_ 40 and −20 ppm. Based on the shape and linewidth of the emergent signal, we believe it belonged to large and mostly amorphic particles, with low molecular mobility and short transverse relaxation times. These changes developed within 2 h and then remained essentially unchanged for hours, except for the fact that the isotropic signals gradually increased [37].

In good accordance with literature data, Trypsin perturbed the structure and function of TMs. It accelerated the decay of ΔA_515_ and gradually diminished its initial amplitude (Appendix A). This might have been caused, at least in part, by the loss of PSII activity [81,82], which explains the diminishment in the 77 K emission bands of PSII relative to those of PSI (Appendix A)—see also [76]. As a consequence of the unstacking of the granum TMs [76,81], Trypsin decreased the (−) 673 nm psi-type band relative to the (+) 685 band (Appendix A) (cf. [74]). In agreement with literature data [81,82], we also observed the gradual diminishment of PSII activity, indicated by the decrease in the F_v_/F_m_ ratio (Appendix A). Further, CN-PAGE revealed characteristic changes in the pattern of PPCs; most prominently, Trypsin disassembled the supercomplexes and monomerized the PSII dimers (Appendix A). 

#### 3.3.2. Proteinase K

Similar to Trypsin, the most prominent effect of Proteinase K was the diminishment in the H_II_ phase; only minor variations were observed in the lamellar and isotropic phases (Figure 9). We also observed the emergence of a broad signal, evidently originating from the PG-containing breakdown product, as seen in the presence of Trypsin. In addition, similar albeit less marked effects were observed in the structural and functional parameters tested (Appendix A). Because of the low activity of this protease at 5 °C, the incubations of TMs, both in the presence or absence of Proteinase K, were performed at 22 °C, while all other measurements were carried out at 5 °C. 

In summary, our experiments with Trypsin and Proteinase K revealed that these proteases essentially eliminated the H_II_ phase of TMs. These data strongly suggest that the participating lipids are associated either with soluble proteins that are attached to the outer surface of TMs or with subunits or polypeptide sections of stromal-side protruding membrane proteins, which might be surrounded or encapsulated by lipid molecules. The large amplitude of the ^31^P-NMR band arising from the lipid phase with inverted hexagonal geometry showed that this phase, in intact TMs, contained large amounts of lipids (~40% of the detectable bulk lipids—see Appendix A). However, as previously tested on granum and stroma TM fractions [38] and on intact TMs (unpublished data), they did not adopt long-range order structures. 

The presently available data do not allow us to identify the protein(s) or polypeptide sections involved in the formation of the H_II_ phase. CURT1 protein, which possesses a Trypsin-sensitive stromal-side domain [68] is a possible candidate. The involvement of CURT1 would imply the presence of a lipid phase with H_II_ geometry in the thylakoid margin. Further candidates include lipocalin enzyme zeaxanthin-epoxidase [72], components of the large stroma-side protruding subunits of PSI [83], and the tightly bound ferredoxin:NADP^+^ oxidoreductase [84,85], as well as the NAD(P)H–dehydrogenase complex [80]. In general, hydrophobic peptides have been shown to be capable of forming peptide-rich H_II_ lipid phases [86,87], and water-soluble proteins can also be encapsulated in different non-bilayer phases [88,89]. In this context, it is interesting to recall that water-soluble respiratory electron-transport chain protein cytochrome c has been shown to promote the formation of the H_II_ phase in model membranes containing cardiolipin, the main non-bilayer lipid of the IMMs [90,91,92].

## 4. Conclusions

Based on our findings, we propose the following assignments of the different lipid phases of TMs: (i) the bilayer phase is ascribed to the protein-rich domains of TMs, which contain the membrane-embedded proteins of the photophysical and photochemical apparatus; (ii) isotropic phases are proposed to originate from regions where membranes fuse together or are branching, key factors in the self-assembly of the TM network, and from lipocalin:lipid associations in the aqueous phases, including photoprotective lipocalin-like luminal enzyme VDE; and (iii) H_II_ phase is evidently given rise by TM lipids surrounding or encapsulating stromal-side protruding protein subunits or polypeptide sections, the nature of which is not yet identified. 

We would like to point out that the presently available data are in harmony with the DEM of TMs, according to which there is a dynamic exchange among the different lipid phases, whereby non-bilayer lipids may safe-guard the homeostasis of TMs and significantly contribute to their structural plasticity [21]. 

Further studies are required to establish the exact origin of different non-bilayer lipid phases in terms of better-defined structural entities and to learn more about their physiological roles. Of particular interest is their possible role in membrane energization and the utilization of the proton motive force. It is an intriguing fact that the relative contributions of non-bilayer lipid phases significantly increase with the elevation in the temperature in the physiological range [18,31], parallel with the increase in the photosynthetic electron transport rate [93,94]. It is interesting to point out that in mitochondrial membranes, the temperature-dependent synthesis of ATP shows a positive correlation with the emergence of a non-bilayer lipid phase [20]. Similar mechanism in TMs would suggest that non-bilayer lipids, possibly via forming sub-compartments and/or via warranting the dense packing of extended protein arrays, play more significant roles in the synthesis of ATP than previously thought.

## Figures and Tables

**Figure 1 cells-11-02681-f001:**
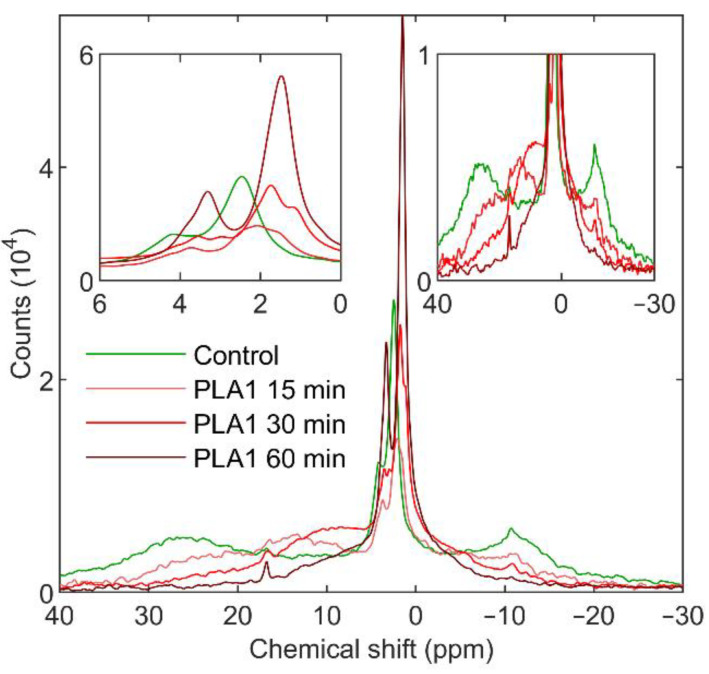
^31^P-NMR spectra of spinach thylakoid membranes; untreated (green) and treated with 2 U mL^−1^ PLA1 for 15 (light red), 30 (red), and 60 (dark red) min. Spectra of three independent batches of isolated thylakoid membranes were averaged; the total acquisition times, at least 30 min for each spectrum; temperature, 5 °C. Insets show the isotropic region (left-hand side) and the lipid phases outside the isotropic region (right-hand side). Spectra were normalized to total Chl contents of 10 mg mL^−1^ and 15 min acquisition times.

**Figure 2 cells-11-02681-f002:**
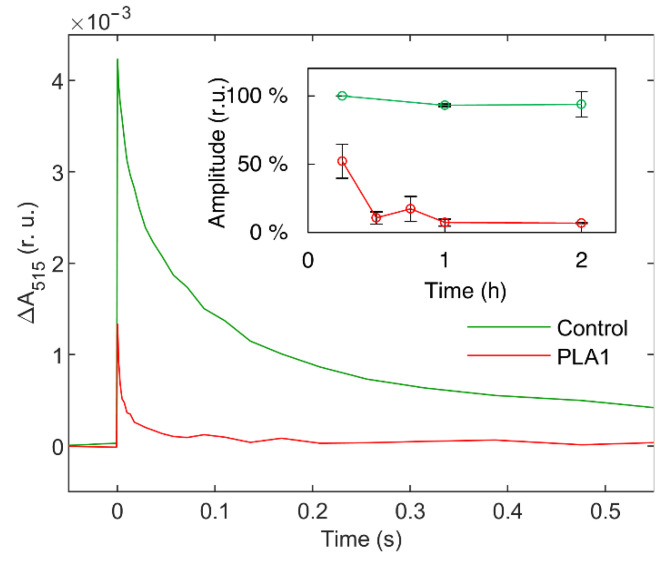
ΔA_515_ electrochromic absorbance shift transients of isolated spinach thylakoid membranes; untreated (green) and treated with 2 U mL^−1^ PLA1 (red) for 15 min. Inset shows the relative initial amplitudes as a function of time; mean values and standard deviations are from three independent experiments.

**Figure 3 cells-11-02681-f003:**
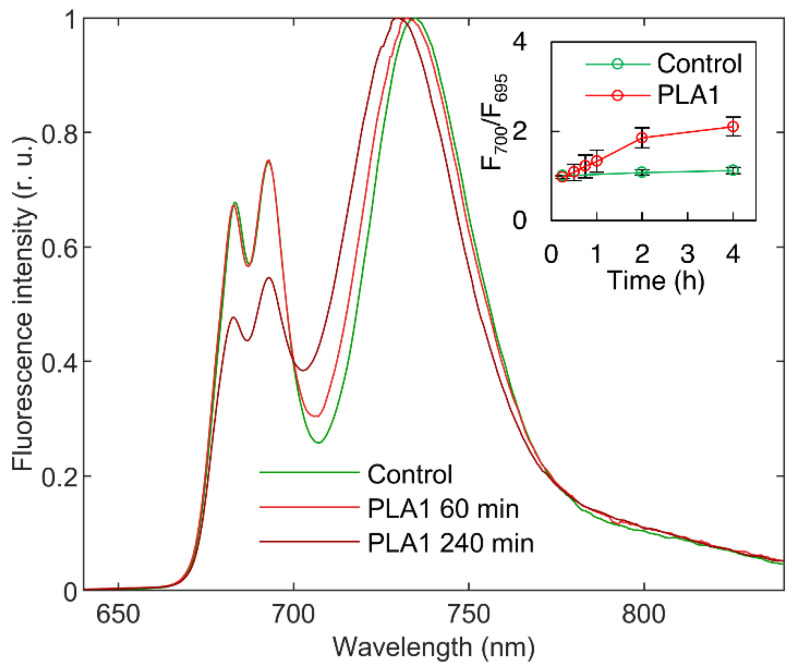
Typical 77 K Chl-*a* fluorescence emission spectra of isolated spinach thylakoid membranes; untreated (green) and treated with 2 U mL^−1^ PLA1 for 60 min (light red) and 240 min (dark red). Excitation wavelength, 475 nm; the spectra were normalized to the maximal intensities of the fluorescence emission. Inset shows the variations in the F_700_/F_695_ ratio as a function of time, relative to the control. Mean values and standard deviations are from three independent experiments.

**Figure 4 cells-11-02681-f004:**
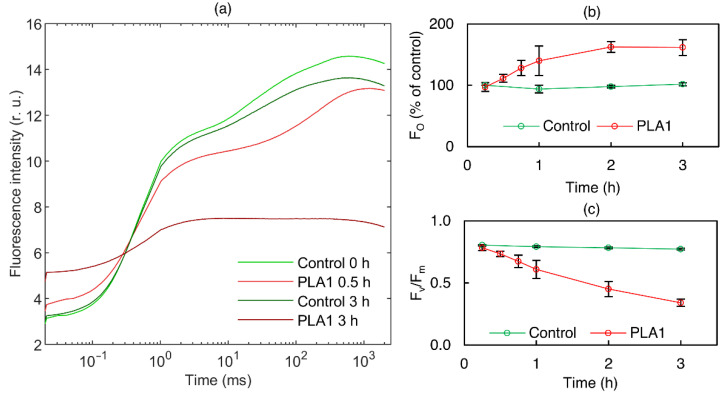
(**a**) Typical fast Chl-*a* fluorescence (OJIP) transients of isolated spinach thylakoid membranes untreated (light green—0 min; dark green—3 h) and treated with 2 U mL^−1^ PLA1 (light red—30 min; dark red—3 h). (**b**,**c**) Time dependences of Chl-*a* fluorescence induction parameters F_o_ and F_v_/F_m_, respectively; untreated (green) and treated with 2 U mL^−1^ PLA1 (red). Mean values and standard deviations are from three independent experiments.

**Figure 5 cells-11-02681-f005:**
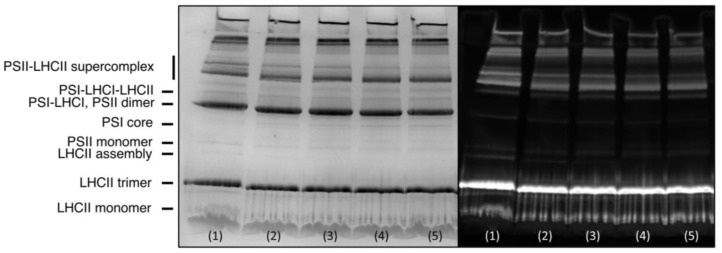
CN-PAGE of untreated isolated thylakoid membranes (lane 1) and membranes treated with 2 U mL^−1^ PLA1 (lanes 2–5). The figure also shows the assignments of the different bands. Lanes 2–5 respectively indicate that the solubilization of samples started 0, 30, 60, and 120 min after PLA1 treatment. Left and right panels respectively indicate the gels transilluminated with visible light and Chl-*a* fluorescence emission (excitation wavelength, 468 nm).

**Figure 6 cells-11-02681-f006:**
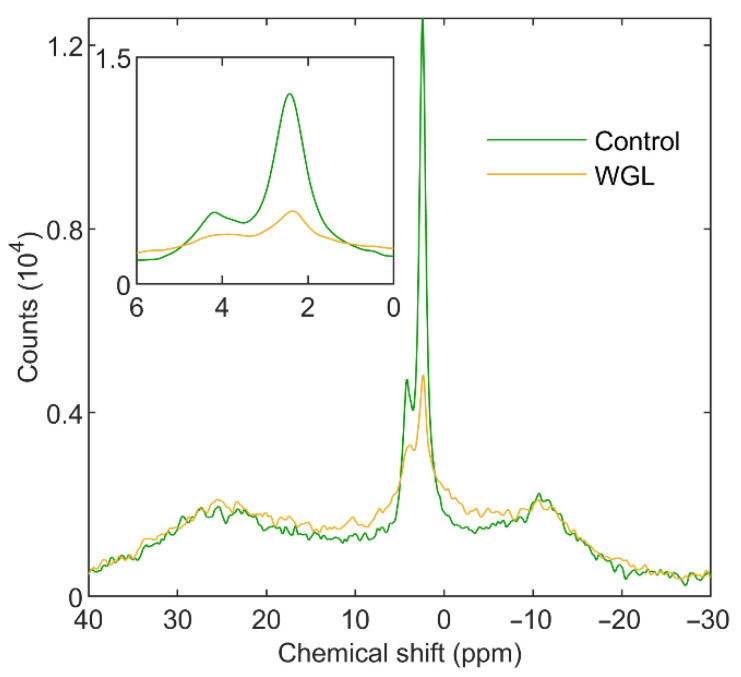
^31^P-NMR spectra of spinach thylakoid membranes untreated (green) and treated with 5, 20, and 100 U mL^−1^ WGL (yellow, brown, and black, respectively). Average of three spectra from independent experiments; each spectrum was recorded within the first hour of treatment; temperature, 5 °C. Inset shows the isotropic region. The spectra were normalized to total Chl contents of 10 mg mL^−1^ and 105 min acquisition times.

**Figure 7 cells-11-02681-f007:**
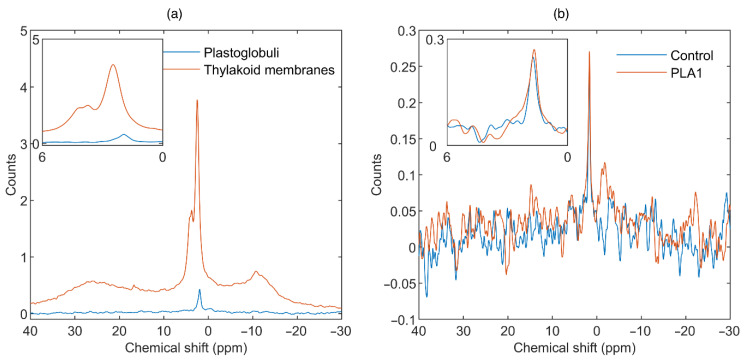
^31^P-NMR spectra of (a) isolated untreated plastoglobuli (blue) and thylakoid membranes (TMs; orange) and (b) untreated plastoglobuli (blue) and plastoglobuli treated for 1 h with 10 U mL^−1^ PLA1 (orange). TM spectrum represents an average from 9 independent measurements with 6.25 h of total acquisition time (AT); plastoglobuli spectra were obtained after 1 h of total AT; all spectra were normalized to 1 h AT. Plastoglobuli and TMs were isolated from equal amounts of mature spinach leaves. Inset shows the isotropic region. Temperature, 5 °C.

**Figure 8 cells-11-02681-f008:**
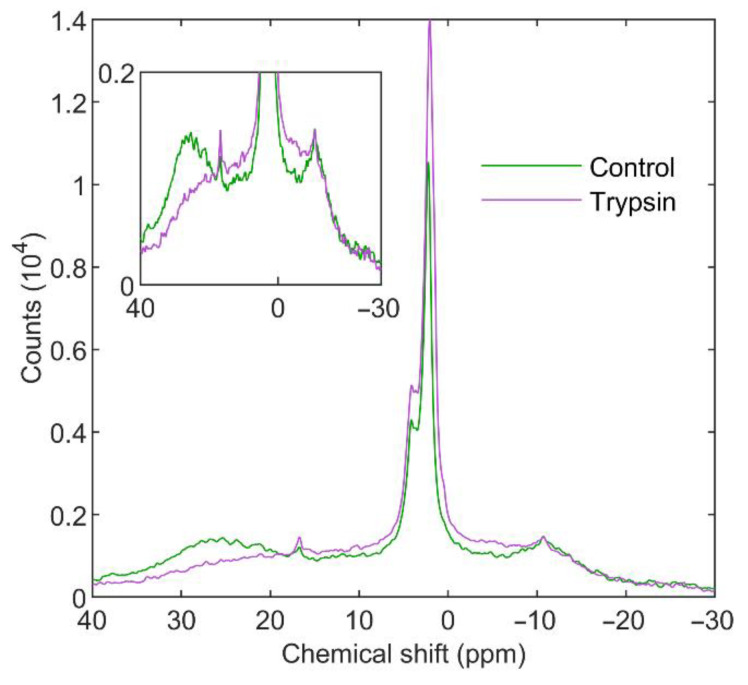
^31^P-NMR spectra of isolated spinach thylakoid membranes untreated (green) and treated with 10 mg mL^−1^ Trypsin (purple) for 2 h. Two spectra from independent experiments of control and treated samples, pairwise displaying similar features, were averaged; temperature, 5 °C. Inset highlights the L and H_II_ regions. Spectra were normalized to total Chl contents of 10 mg mL^−1^ and acquisition times of 75 min.

**Figure 9 cells-11-02681-f009:**
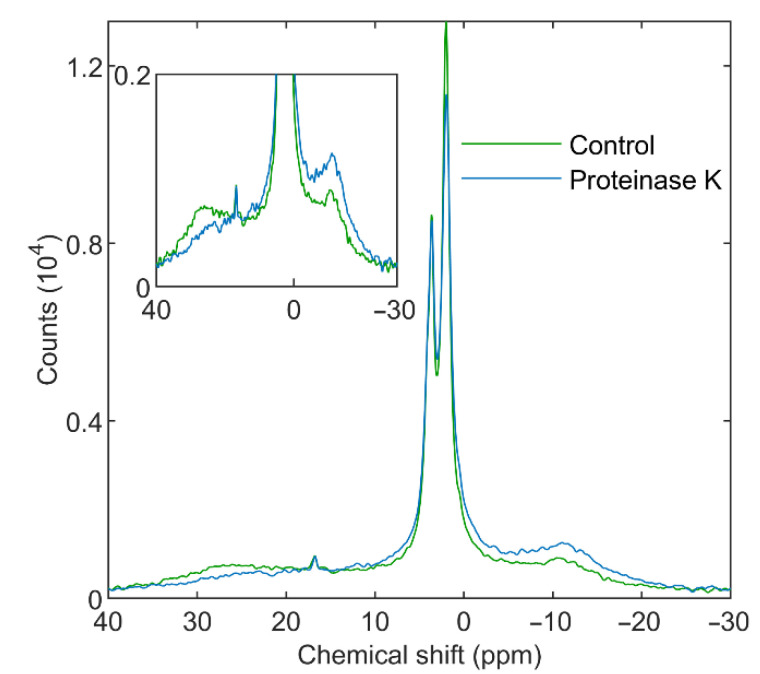
^31^P-NMR spectra of isolated spinach thylakoid membranes; untreated (green) and treated with 16 U mL^−1^ Proteinase K (blue) for 30 min at 22 °C and recorded at 5 °C. Two spectra from independent experiments of control and treated samples, pairwise displaying similar spectral features, were averaged. Inset highlights the L and H_II_ regions. The spectra were normalized to total Chl contents of 10 mg mL^−1^ and acquisition times of 105 min.

## Data Availability

The original data were recorded at Slovenian NMR Center. Processed and derived data are available from the corresponding author (G.G.) upon request.

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
