# Peer review of "Structural Entities Associated with Different Lipid Phases of Plant Thylakoid Membranes—Selective Susceptibilities to Different Lipases and Proteases"

_cells, 2022, doi:10.3390/cells11172681_

Round 1
Reviewer 1 Report
Dear authors:
The paper is interesting, adresses an important topic in which is not frequent to find papers and gives new insights. Even though, there are some points that need to be adressed, all related to the way in which results are presented.
1.- Introduction is very long. It looks like a review rather than a contextualization of the study. Please shorten it to give precise information on the background of the study.
2.- Figure 4b and 4c. There are only error bars in one of the data series. Is there any explanation for this? If error bars are lower than the point width, authors should eliminate the point and use only the line to make error bars visible.
3.- There is no discussion, but after the experiments there is like a kind of summary (i.e. line 413-417 or line 531-554). Is better to compile all these information in a separate discussion to give a general overview of the results and facilitate the understanding of the work and the interpretation of the results.
4- The conclusions are too long. A conclusion should be a single paragraph summarizing the main findings, not a short discussion. Please consider to write a real discusion using this information. In the conclusion just state the main findings and the significance of your work.
Reviewer 2 Report
The article in question is interesting. It is written in a logical and orderly manner.
However, before publishing it, I suggest you improve a few elements.
1. Too many self-citations.
2. I suggest that you read the manuscript carefully again to eliminate linguistic errors.
3. The conclusions should be concise and relate to the studies discussed in the manuscript. I suggest shortening them a bit and writing them so that they mainly concern the research included in the discussed work and possible plans of these research for the future.
